# Long-term nitrogen burial exceeds denitrification in global fjords

Henry L. S. Cheung [1] ✉, Lubrina S. Levin[2], Craig Smeaton [3], Tobia Politi [1], Bo Thamdrup [2], Isaac R. Santos [1] & Stefano Bonaglia [1]

Nitrogen (N) availability regulates primary productivity and hence directly affects global oceanic carbon sequestration. Global fjords account for up to 11% of marine carbon burial. However, N loss via sediment burial remains largely unquantified. Here, we show that global fjords are hotspots of N burial, accounting for up to 18% of oceanic N burial despite only covering 0.1% of the ocean area. Burial is the dominant N loss mechanism, exceeding microbial N loss via denitrification and anammox, which are generally considered the major N loss mechanisms. Microbial N loss dominates in anoxic fjords and appears to be a function of temperature and nutrient availability. Overall, fjords efficiently sequester excess N in sediments over long time scales. Accelerated warming will promote both N burial from increased primary production and microbial N loss from warmer temperatures, affecting N budgets in fjords and in the ocean in general.

Nitrogen (N) availability is intimately linked to carbon held within marine organic matter[1,2]. In addition to typical blue carbon habitats such as mangroves, seagrass meadows, and salt marshes, fjords are also key coastal marine ecosystems, burying ~11% of the global marine organic carbon[3,4]. Since N availability is a key driver of marine primary productivity, N uptake and loss are intimately linked to the efficiency of carbon sequestration[5]. Growing anthropogenic N inputs to coastal oceans have promoted eutrophication and subsequent oxygen ($O_2$) depletion[6]. Additionally, warming-induced stratification reduces $O_2$ supply to fjord deep waters, thereby favouring deoxygenation. Coastal N enrichments not only modify carbon sequestration efficiency[7,8], but also enhance greenhouse gas emissions, which offset the climate-mitigating capacity of coastal systems[9,10]. Quantification of N loss in mid- to high-latitude blue carbon ecosystems is hence critical to understanding their role in mitigating N enrichment from both direct (agricultural runoff, aquaculture, and wastewater) and indirect (glacial meltwater and soil erosion) sources.

Fjords are mid- to high-latitude coastal ecosystems with steep catchments and restricted deep water movement. Such features result in high sedimentation and organic matter burial[11]. Despite numerous studies on carbon cycling in global fjords[3,12], less is known about N cycling processes, including burial rates of blue nitrogen–N

sequestered in marine sediments. Excessive N in fjords can be removed via microbial $N_2$ production as well as burial of particulate organic N and mineral-bound inorganic N. Denitrification, the microbial reduction of nitrate to $N_2$, is often considered the main N removal mechanism in the ocean[13,14]. As a consequence, earlier fjord studies have mainly focused on N loss via denitrification[15,16]. The contribution of global fjords to marine N burial[17] remains largely unknown. In addition, the overall importance of anammox–the second most important microbial $N_2$ production pathway[18]–remains unclear in fjord systems.

Fjord N loss can be directly and indirectly related to anthropogenic stressors. Weak mixing limits $O_2$ replenishment in fjord deep waters, which may lead to hypoxic or even anoxic conditions[19]. Deoxygenation has been spreading in fjords worldwide, stimulating microbial N loss[20–23]. As a consequence of increased anthropogenic N loads to the ocean, N inventories in global fjords have increased over the past decades[24]. Anthropogenic activities also indirectly impact N cycles in remote, polar fjords[25]. Arctic atlantification, i.e., the extension of warm, nutrient-rich Atlantic waters into the Arctic[26], amplifies warming in the Arctic Ocean[27], favouring deoxygenation[28] and thereby altering microbial N transformation[29]. Accelerating glacier melting influences fjord biogeochemistry via the introduction of N-rich

[1]Department of Marine Sciences, University of Gothenburg, Gothenburg, Sweden. [2]Nordcee, Department of Biology, University of Southern Denmark, Odense M, Denmark. [3]School of Geography and Sustainable Development, University of St Andrews, St Andrews, UK. ✉e-mail: lok.shan.cheung@gu.se

meltwater[30,31]. It is thus essential to resolve key N loss mechanisms to understand N cycling in global fjords under increasing anthropogenic pressures.

Here, we show that N burial is the main N loss pathway in fjords worldwide by combining original measurements from five fjords in Sweden and Iceland with a global compilation of data from 74 fjords. N burial rates were derived using a compiled database of 121 sites worldwide, each with measured mass accumulation rates and sediment N content. The contribution of N burial to total N loss was then contrasted to empirical rates of microbial $N_2$ production via denitrification and anammox. We used a weighted bootstrap analysis that corrects for sampling biases to upscale global N burial and $N_2$ production rates[32]. Finally, we show that $N_2$ production is primarily stimulated by $O_2$-depletion, increasing water temperature, and inorganic N contents. Ongoing anthropogenic impacts will thus progressively promote the dominance of microbial N loss in fjords globally.

## Results and discussion

### Large nitrogen burial rates in global fjords

Nitrogen burial, represented as nitrogen accumulation rates ($N_{AR}$), ranged from 0.10 to 16 g N m$^{-2}$ yr$^{-1}$ in global fjords (Fig. 1a). Our spatial weighted bootstrap analysis yielded a global $N_{AR}$ median (interquartile range, IQR) of 3.9 (3.7 – 4.1) g N m$^{-2}$ yr$^{-1}$ (Fig. 2a). This areal rate exceeds most marine systems and even global lakes (2.5 g N m$^{-2}$ yr$^{-1}$)[33], implying that fjords are hotspots for N burial. Arctic, North Atlantic, and North Pacific fjords exhibited N burial rates 14 to 19% higher than the global median, whereas those in the South Pacific (Chile and New Zealand) and Antarctica have rates 45 to 85% lower than the median (Fig. 2a). Fjord N burial rates significantly increased with latitude (Fig. S1). Large N burial rates were mostly observed in high-latitude fjords ($\geq 60°$) in Greenland, Canadian Arctic, and Svalbard (Fig. S2). Scaling to the updated global fjord area (259,899 km$^2$; ref. 34), we estimated a median (IQR) global N burial of 1.0 (1.0 – 1.1) Tg N yr$^{-1}$ (Table S1), representing 2–10% of N burial in global ocean sediments (10–58 Tg N yr$^{-1}$; ref. 17. and references therein). Using the larger, commonly referred fjord area (455,535 km$^2$; ref. 35) increases estimate to median of 1.8 (1.7–1.8) Tg N yr$^{-1}$, or 3–18% of global marine N burial[17]. Despite only representing 0.1% of the global ocean area[34,36], such disproportionately high N burial rates reveal that fjords are hotspots for N burial in the ocean.

Since N burial rates are derived from mass accumulation rates ($M_{AR}$), the high N burial capacity is necessarily linked to $M_{AR}$ (Fig. 3a). Similar to C burial, N burial rates are predominantly associated with $M_{AR}$[3,37,38]. N and C burial were highly correlated across global fjords (Fig. S1). High $M_{AR}$ was often observed in polar and subpolar regions (Fig. 2b). The global pattern of $M_{AR}$ generally followed that of N burial across ocean basins, except for the two glacier-influenced Antarctic fjords, where $M_{AR}$ was comparable to the global median yet with lower $N_{AR}$ (Fig. 2a, b). Higher $N_{AR}$ were observed at the head of polar fjords (e.g., Hornsund and Kongsfjorden) (Fig. S2) with marine-terminating glaciers acting as large allochthonous sediment sources[5,37,39]. In polar fjords, large sediment inputs from glacier meltwater and erosion[11] can act as key sources of fjord inorganic N[40], which could consequently sustain high N burial rates found in Arctic fjords and Greenland fjords in the North Atlantic ($>60°N$).

Sediment stoichiometry revealed latitudinal variations in the quality of buried organic matter, as indicated by the C:N ratio, and N burial (Fig. S1). The negative correlation between this ratio and N burial implies higher N burial rates in fjords with high labile organic matter. A clear shift from preponderantly terrestrial (refractory) to marine (labile) organic matter in sediments occurred from temperate towards polar fjords (Fig. S1). Labile organic matter from marine primary production has a lower C:N ratio ($<10$) than terrestrial organic matter ($>15$)[41,42]. This aligns with the low sediment C:N ratios in polar fjords, where organic matter is predominantly derived from marine primary

production[5]. Notably, high productivity can be sustained by nutrient inputs from nutrient-rich deep water upwelling, subglacial discharge, and/or N-rich meltwater from retreating glaciers[43–45]. High sedimentation rates shorten $O_2$ exposure for organic matter degradation, allowing efficient preservation of labile organic matter[46] and associated N burial in Arctic fjords[47]. Combined with high sedimentation rates from retreating calving glaciers[48], an even higher N burial is currently occurring in high-latitude fjords. With increasing sedimentation and decreasing organic matter C:N ratio (Fig. 3a), N burial in fjords is also expected to increase. In contrast, more refractory organic matter in temperate, terrestrial-dominated fjords (Fig. S1) reflects the underlying oligotrophic conditions with low primary production[2], and associated lower $M_{AR}$ and N burial, particularly in the Southern Hemisphere (Fig. 2a, b).

### Fjord nitrogen burial surpasses microbial nitrogen loss by denitrification and anammox

Denitrification (nitrate → $N_2$) and anammox (ammonium + nitrite → $N_2$) are often considered the dominant N loss pathways in marine ecosystems[49]. Nitrogen loss by these two $N_2$ production processes ranged from 0.10 to 21 g N m$^{-2}$ yr$^{-1}$ across 20 fjords globally, including our observations from five fjords (Fig. 1b). Spatially weighted bootstrap analysis yielded a global median (IQR) of 2.4 (2.0–2.8) g N m$^{-2}$ yr$^{-1}$. Fjord $N_2$ production decreased at high latitudes and increased in temperate and subpolar fjords ($<60°$) in the North Atlantic (Fig. 1a). The lower regional median (~0.9 g N m$^{-2}$ yr$^{-1}$) in Greenland and Svalbard likely reflects lower temperatures and organic content availability for $N_2$ production processes compared to temperate fjords, given that both temperature and organic carbon were positively correlated with sediment $N_2$ production (Fig. S1). Overall, water temperature, nitrate + nitrite concentrations, and oxygen concentrations are significant factors driving fjord $N_2$ production (Fig. 3b). While based on measurements from only two anoxic fjords, $N_2$ production was notably higher under anoxic conditions (Fig. 4a). Median $N_2$ production rates in anoxic ($<1 \mu M$ $O_2$) fjords is 9 times higher than in oxic fjords (Fig. 4). Hence, anoxic fjords, in particular, are hotspots of $N_2$ production at rates up to 6 times higher than the global fjord median.

Denitrification and anammox primarily occur in anoxic sediment layers, where fixed N derives mostly from nitrification or diffusion from the overlying water. Benthic denitrification increases with decreasing $O_2$, particularly below the typical threshold of hypoxia ($<63 \mu M$; ref. 50.). For instance, greater sediment denitrification and anammox rates were measured when bottom water $O_2$ concentrations decreased from 260 to 61 $\mu M$ in Gullmar Fjord, Sweden (Fig. S3). Similarly, greater $N_2$ production was observed in the hypoxic Loch Etive[51] compared to oxic fjords (Fig. S4). When anoxia develops in fjord basin waters, benthic $N_2$ production will cease due to restricted nitrification and limited nitrate concentration in deep water[52]. The active nitrate reduction zone then rises from the sediment to the water column[53]. Typically, this layer's thickness increases orders of magnitude from millimetres-centimetres in sediments[54] to decimetres-meters scale in the water column oxic-anoxic interface[21,22]. Such substantial increases in the volume of the nitrate reduction zone enhance overall $N_2$ production in anoxic fjords.

Including our measurements from five fjords, paired quantifications of N burial and $N_2$ production become available for a total of 16 fjords worldwide (Fig. 1c). N burial exceeded microbial $N_2$ production as the major N loss mechanism in 75% of these fjords. The median N burial contribution to total N loss (N burial + $N_2$ production) was 65% across fjords (Fig. S4). High-latitude fjords with large sedimentation rates and low $N_2$ production rates in Greenland and Svalbard exhibit remarkably high N burial contributions reaching 90% (e.g., Kobbefjord). In contrast, microbial $N_2$ production remarkably exceeded burial in three anoxic/hypoxic fjords (Saanich Inlet, By Fjord, and Loch Etive) (Fig. S4). However, even including exceptionally high $N_2$ production

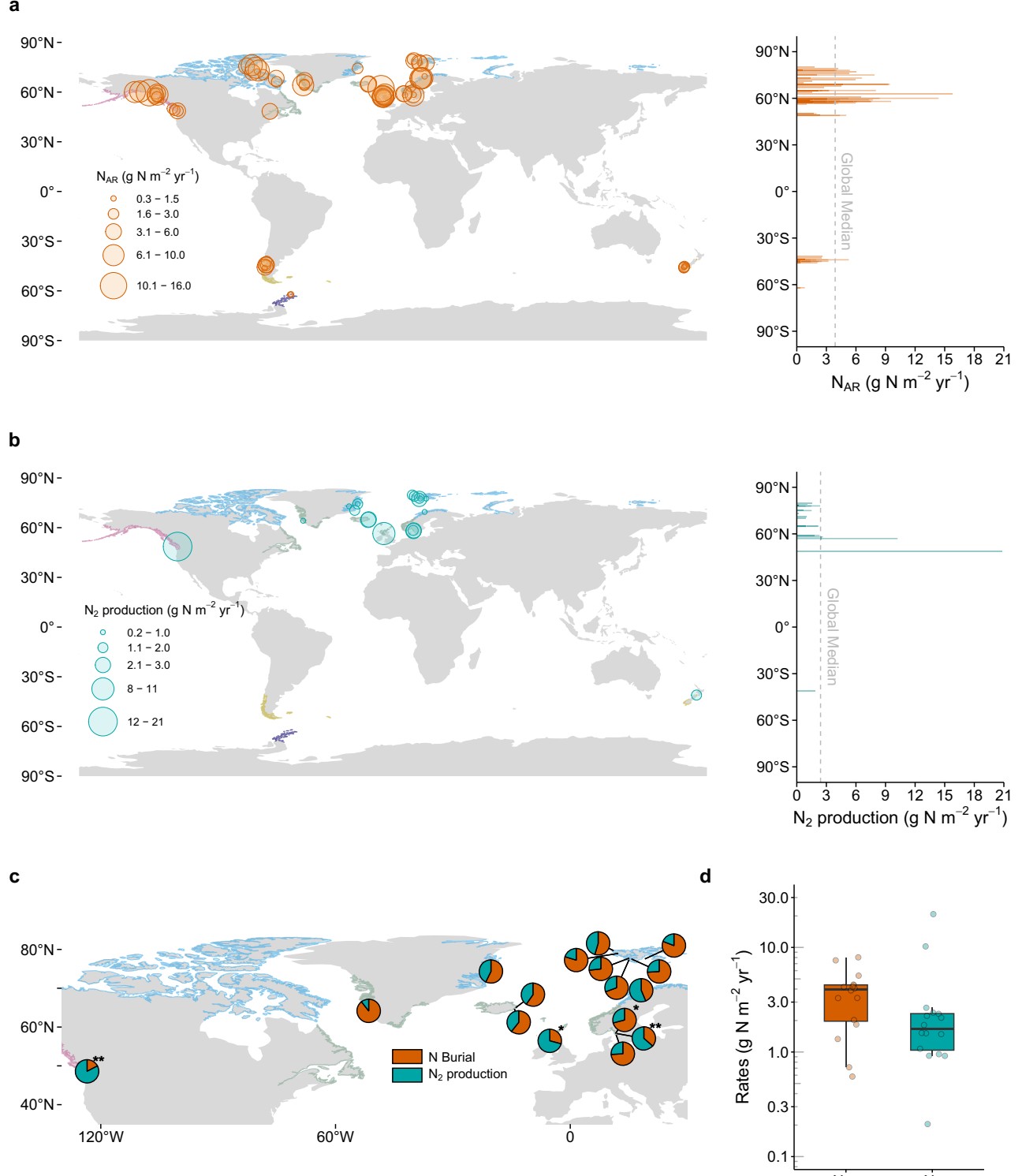

**Fig. 1 | Global distribution of fjord nitrogen burial and $N_2$ production rates.**
Global distribution of fjord (**a**) sediment nitrogen burial ($n = 79$) and (**b**) sediment or water column $N_2$ production rates ($n = 20$). Bolded, colour-coded coastlines indicate fjord regions. Bars indicate rates at the corresponding latitude. **c** Contribution of sediment burial (orange) and $N_2$ production (teal) to total nitrogen loss (sediment burial + $N_2$ production) in fjords where both processes were measured, and * and ** denote hypoxic and anoxic fjords, respectively. **d** Rates of sediment burial ($N_{AR}$) and $N_2$ production ($N_2$) in fjords globally.

hotspots in anoxic/hypoxic fjords, these global estimates show that N burial exceeds $N_2$ production (Fig. 4c), challenging general paradigms for N removal in the marine environment[55–57]. Our analysis clearly indicates the importance of N burial as a long-term N storage mechanism in fjords globally.

**Fate of nitrogen in deoxygenated, eutrophic and warmer fjords**
Up to nine-fold greater $N_2$ production in anoxic than in oxic fjords (Fig. 4a) implies that complete deoxygenation shifts the dominant N removal pathway from sedimentary burial toward microbially mediated N loss. Fjord $N_2$ production increased gradually from oxic to

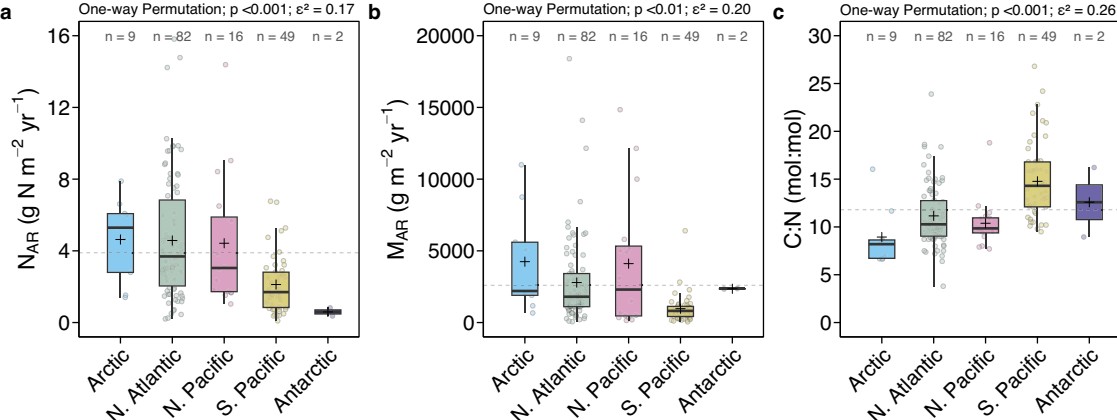

**Fig. 2 | Distribution of accumulation rates and stoichiometry in fjord sediments across ocean basins.** The range and variability of sediment (**a**) nitrogen accumulation rates ($N_{AR}$), (**b**) mass accumulation rates ($M_{AR}$), and (**c**) sediment carbon to nitrogen molar ratio (C:N) across ocean basins (colour-coded). Dashed horizontal line indicates bootstrapped global median value. Crosses indicate bootstrapped medians of the corresponding region. Differences among ocean basins were assessed using permutation-based one-way tests, with global $p$ values reported. Effect sizes were quantified using epsilon-squared ($\varepsilon^2$) derived from the Kruskal–Wallis statistic.

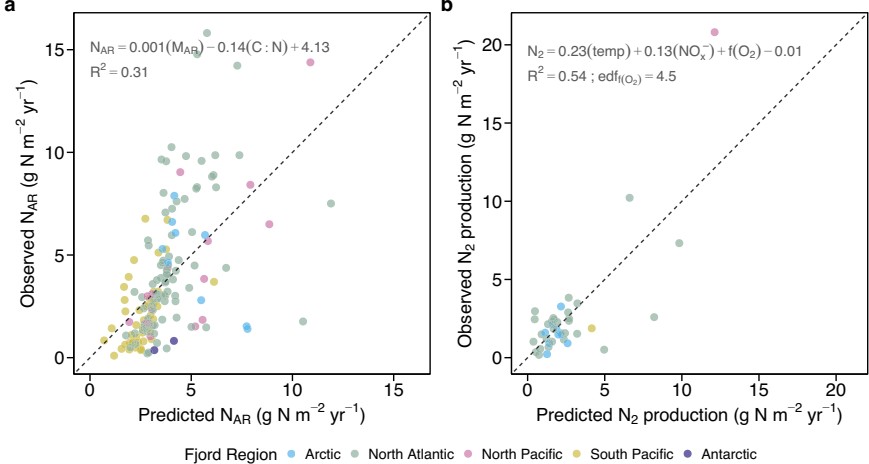

**Fig. 3 | Predictive models for sediment nitrogen burial and $N_2$ production across fjords globally. a** Observed versus predicted fjord nitrogen accumulation rates ($N_{AR}$) derived from a generalized linear model including sediment mass accumulation rates ($M_{AR}$) and sediment C:N ratio (C:N). **b** Observed versus predicted fjord $N_2$ production rates derived from a generalized linear model including water temperature (temp), nitrate and nitrite concentrations ($NO_x^-$), and oxygen concentrations ($O_2$) fitted using restricted maximum likelihood, with effective degrees of freedom for the fit ($edf_{fO2}$) reported. Data points are colour-coded on the basis of fjord regions and the dashed line indicates the 1:1 relationship. Model coefficients and goodness-of-fit ($R^2$) of the corresponding model are shown.

hypoxic conditions with a first threshold at 99 μM $O_2$, followed by a sharp increase as $O_2$ declined toward anoxia down to low micromolar range ( < 20 μM) (Fig. S5; Table S2). Remarkably, $N_2$ production rates in anoxic fjords were comparable to or even higher than those in the anoxic, highly eutrophic Baltic Sea (0.30 to 11 g N m$^{-2}$ yr$^{-1}$)[58]. The substantial increase in average $N_2$ production in anoxic fjords is primarily due to the regime shift and expansion of the nitrate reduction zone, i.e., the anoxic but nitrate containing zone[58] (Fig. 5). Indeed, anoxia can inhibit nitrification and subsequentially limit $N_2$ production, resulting in a transient phase of elevated $N_2$ production between oxic and anoxic conditions. Since no significant differences in N burial were found between oxic and anoxic fjords (Fig. 4b) we propose that $N_2$ production will surpass N burial and dominate N loss with developing anoxia.

Combining the two mechanisms, median (IQR) total N loss was 5.3 (3.5–6.1) g N m$^{-2}$ yr$^{-1}$ in oxic fjords and increased to 6.2 – 14.0 and to 3.6 – 25.0 g N m$^{-2}$ yr$^{-1}$ in hypoxic and anoxic fjords, respectively. The extent of future changes in fjord N loss pathways are intrinsically sensitive to perturbations of $O_2$ levels, temperature and nutrient availability (Fig. 3b), which are all intimately linked to a warming climate[59]. In particular, intensified warming in the Arctic, or Arctic amplification, will stimulate deoxygenation and will likely create hypoxic zones in narrow and inner coastal basins[28]. Hence, we expect a stronger contribution of $N_2$ production to N loss in fjords as currently observed in hypoxic and anoxic fjords worldwide (Fig. 4c). Ocean warming also promotes the northward expansion of warmer, nutrient-rich waters from the Atlantic into the Arctic[60–62]. Warmer fjord waters favour sediment $N_2$ production (Fig. S1)[63]. Higher inorganic N availability (for example from increased Atlantic inflow[64]) can also stimulate sediment $N_2$ production (Fig. 3b), particularly via anammox[65]. In addition to the direct stimulatory effects of warming and nutrient enrichment onto $N_2$ production (Fig. 3b), the increased availability of labile organic matter due to atlantification-driven primary production may increase sediment $N_2$ production in Arctic fjords[45,63,66]. Climate change stimulates primary production in high-latitude fjords and subsequent deposition of labile organic matter on sediments[45], favouring both N burial and benthic $N_2$ production (Fig. S1). Combined

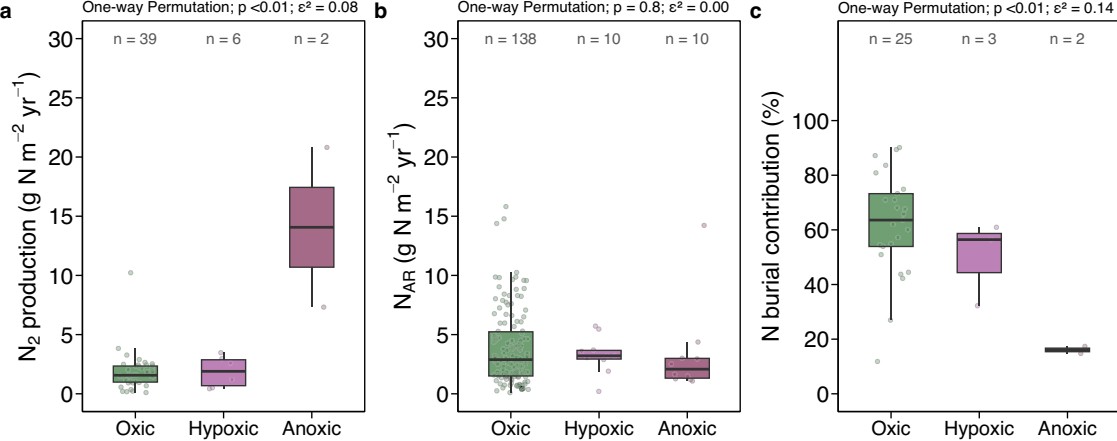

**Fig. 4 | Rates of N loss in fjords under different redox conditions. a** $N_2$ production and (**b**) nitrogen accumulation rates ($N_{AR}$) in anoxic, hypoxic, and oxic fjords. **c** Contribution of N burial to total fjord N loss (sum of $N_2$ production and N burial) under each redox condition. Global differences among redox conditions were assessed using permutation-based one-way tests, with $p$ values reported. Effect sizes were quantified using epsilon-squared ($\varepsilon^2$).

with potentially higher N burial due to warming-mediated glacial retreat and increased sedimentation rates[48], our results suggest an increased N loss in global fjords along with a warmer and less oxygenated ocean.

Our global analysis demonstrates that sediment burial is the most effective N sink mechanism in fjord systems under current conditions. Despite covering <0.1% of the ocean, fjords contribute up to 18% of global N burial. This makes them hotspots of blue nitrogen, i.e., nitrogen sequestered in marine sediments. On average across fjords globally, sediment burial accounts for $60.6 \pm 5.0\%$ of total N loss, exceeding denitrification ($34.8 \pm 4.5\%$) and anammox ($4.6 \pm 2.0\%$). Thus, burial plays a surprisingly significant role in the long-term N removal of fjords globally, making it a crucial mechanism for mitigating coastal N enrichment at medium and high latitudes. Additionally, N burial is an effective removal pathway with minimum climatic impact, compared to microbial nitrification and denitrification, which produce the greenhouse gas nitrous oxide ($N_2O$; equivalent to ~1% of $N_2$ production in an anoxic fjord; ref. 21,67.). Burial in fjords is driven by sedimentation rates and the higher lability of deposited organic particles (Fig. 3a), which can be favoured by increasing rates of primary production, particle deposition, and glacial retreat[68,69]. However, natural hazards such as landslides and floods can alter sediment delivery and hence N burial efficiency[70,71].

The control of environmental factors on fjord microbial N loss via denitrification and anammox is intricate[72]. At the regional scale, warming and increasing N concentrations will enhance microbial N loss following expanding deoxygenated areas[24,28]. Given the hydrogeomorphic characteristics of fjords and the warming climate, $O_2$ deficiency is expected to develop more frequently, including high-latitude fjords impacted by Arctic amplification and atlantification[26,59]. Indeed, we show that microbial N loss dominates over geological burial loss due to the shift in $N_2$ production regimes following the development of anoxia (Fig. 5). Rates of microbial N loss are governed by redox conditions, substrate availability, and the extent of the active nitrate reduction zone. Despite large rate variability, our observations establish a robust link between fjord N loss and deoxygenation. Additional work should focus on the climatic effect of fjord N loss via $N_2O$ production, and on refining the role of fjords in the global N cycle. Whether N burial can counter coastal nutrient overenrichment (i.e., eutrophication) and curb greenhouse gas emissions largely depends on coastal nutrient management. The future partitioning of N loss mechanisms in fjords is predominantly linked to their deoxygenation level, eutrophic status, and hydrological response to global warming.

## Method

### Global fjord database collection and upscaling

Our global fjord database combining sediment mass accumulation rates and N content, consists of 158 observations across 79 fjords. Major fjord regions are covered[11], including North-western Europe (Scotland, Sweden, Norway, Iceland, and Faroe Islands), Greenland, Svalbard, Western Canada, Eastern Canada, the Canadian Arctic archipelagos, Alaska, Patagonia, New Zealand, and Antarctica. We excluded shallow fjards (which are not true fjords but rather drowned estuaries) with average water depth <10 meters, such as those in Denmark, due to the hydrodynamical and physicochemical differences. We assumed that sediment particle size was relatively homogenous across sites, given a mean porosity of $0.78 \pm 0.06$. Global rates were determined by upscaling to the global fjord area[34,35] (Table S1).

Nitrogen burial rates, expressed as $N_{AR}$, were further compared to microbial N loss rates (i.e., $N_2$ production) that were measured in the corresponding fjords. Our global compilation of fjord $N_2$ production measurements consists of 47 observations across 20 fjords. Reported areal rates were measured by either intact core incubations, water column incubations, or in situ chamber incubations using isotope pairing techniques[73,74]. Areal $N_2$ production rates obtained in the water column were obtained in the case of anoxic fjords. Measurements of both $N_2$ production and N burial are available for a total of 16 fjords located mainly in NW Europe, Svalbard, Greenland, and Eastern Canada.

### Original observations in Swedish and Icelandic fjords

Data on both N burial and $N_2$ production were collected in five temperate (57–62°N) and subpolar (65°N) fjords along the west coast of Sweden and the east coast of Iceland, respectively (Fig. S6). These fjords have distinct watershed characteristics, anthropogenic influences, and redox gradients, including oxic, seasonally hypoxic, and long-term anoxic conditions (Table S3). Samples were collected at three sampling stations at the head, centre, and mouth of each fjord aboard the R/V Skagerrak. For the Swedish fjords, samples for mass accumulation rates and sediment stoichiometric analysis were collected in September-October 2021[75] and $N_2$ production rates in sediments and water columns were determined by $^{15}N$-labelling incubation experiments in 2023 and 2024. For Icelandic fjords, samples for all analyses and incubations were collected in June 2023. Mass accumulation rates and sediment stoichiometry from Icelandic fjords were reported in a previous publication[76], which were

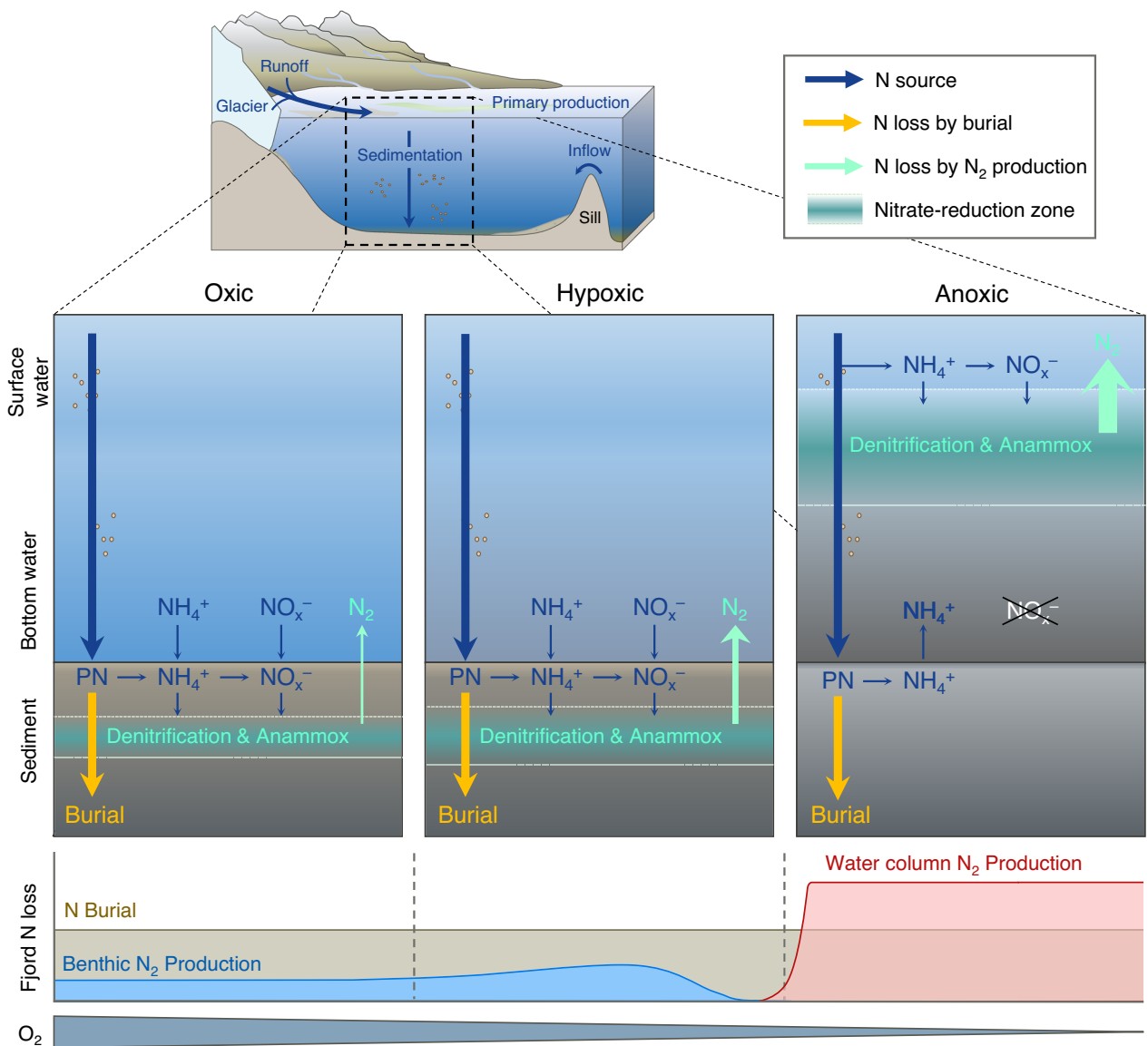

**Fig. 5 | Conceptual model illustrating the transition of fjord nitrogen loss rates as a response to deoxygenation.** Nitrogen (N) sources in both particulate and dissolved form are introduced to the fjord via terrestrial runoff, glacial melting, and oceanic inflow. Dissolved N is assimilated by primary production (or binds with minerals) and is subsequently deposited onto sediments as particulate nitrogen (PN) via sedimentation. Fractions of the deposited PN are then remineralised, nitrified and utilised for microbial $N_2$ production, while a conspicuous remaining fraction is buried in the deep layer. Under oxic conditions ($>100\,\mu M\ O_2$), N burial surpasses benthic $N_2$ production, and water column $N_2$ production is not active. We hypothesize that N loss via sediment burial remains consistent despite the bottom water may experience occasional deoxygenation. Benthic $N_2$ production peaks at typical hypoxic thresholds ($63\,\mu M\ O_2$) as low $O_2$ levels stimulate microbial $N_2$ production, yet it diminishes when approaching anoxia. Water column $N_2$ production dominates fjord N loss when anoxia (defined as $<1\,\mu M\ O_2$) develops in bottom water.

determined based on particle-reactive radiotracer-derived age models and sediment stoichiometry. Detailed descriptions of sampling procedures of each campaign are given in the Supplementary Materials.

## Determination of nitrogen accumulation rates and $N_2$ production

Total N burial rate, defined as N accumulation rate ($N_{AR}$) in the sediment, was calculated from mass accumulation rates, porosity, bulk density, and sediment N content in collected sediment cores and in the literature (Supplementary Materials and references therein). Mass accumulation rates ($M_{AR}$) were determined based on particle-reactive radiotracer-derived age models in the sediment cores and were obtained from published data[75]. Age models were created for each core using $^{210}Pb$, $^{137}Cs$ and/or $^{241}Am$ profiles. Dry bulk density and porosity were determined for each sediment slice by measured wet and dry weights. Radionuclides $^{210}Pb$, $^{137}Cs$, and $^{241}Am$ were analysed using an ORTEC HPGe GWL-series well-type coaxial low-background germanium detector. Linear sedimentation rates were estimated by the constant flux–constant sedimentation (CF-CS) model, ensuring a logarithmic regression fit with $R^2 > 0.75$, and were further constrained by $^{137}Cs$ and $^{241}Am$ peaks. Mass sediment accumulation rates were calculated using the constant rate of supply model and validated against $^{137}Cs$ peaks. N accumulation rates were derived from multiplying $M_{AR}$ by the weight fraction of total nitrogen (TN%). When only organic carbon accumulation rates and sediment C:N ratio were reported in the literature, $N_{AR}$ were calculated considering the sediment stoichiometric ratio.

Sediment $N_2$ production rates were determined through $^{15}N$-labelling incubation experiments by the revised-isotope pairing

technique[74]. Collected intact cores were spiked with $^{15}NO_3^-$ (11, 24, 56, and 120 μM mean final concentration) to determine rates of $N_2$ production (denitrification and anammox) and N recycling (dissimilatory nitrate reduction to ammonium rates, DNRA). At the centre station of By Fjord, total $N_2$ production rates were determined in the water column as $N_2$ production is restricted in the sediment due to depletion of $O_2$, $NO_x^-$ and any other N oxide species[23]. $^{15}N$-labelling incubation experiments were conducted with samples collected 21 m below the surface, which covered the $NO_x^-$ containing oxic-anoxic water layer. Additional incubation procedures and rate calculations are given in the Supplementary Materials.

## Spatial weighted bootstrap analysis

We performed spatial weighted bootstrap analysis to minimize the impact of geographical sampling bias on our estimates of the global mean of N loss processes. Such an analysis helps to minimise uncertainties derived from small datasets with geographical sampling biases. We assigned sample weights based on their spatial proximity to other samples using the inverse weighting algorithm[32,77] with a spatial scale of ~55 km (0.5 degree). Using the resulting weights (Fig. S7), we performed a bootstrap analysis to generate distributions of 1000 weighted bootstrapped means for both $N_{AR}$ and $N_2$ production rates. The observed raw data distribution and bootstrapped data used in the analyses were illustrated in histograms (Fig. S8). Both spatial weighing and bootstrapping analysis were performed in R version 4.3.2.

## Statistical analysis

The relationship between environmental variability and N removal rates was estimated from the correlation of N burial vs. total $N_2$ production rates with individual environmental factors. Spearman's rank correlation coefficients were calculated to minimize impacts of collinearity. A one-way permutation test was used to reveal the difference of $N_{AR}$ and $N_2$ production rates across oxic and deoxygenated fjords, as well as the difference in mean $N_2$ production processes between oxic and hypoxic sediment in Gullmar Fjord. To examine potential change points in the relationship between $N_2$ production and $O_2$ concentration across redox regimes, we fitted a Bayesian segmented regression model with three linear segments using the mcp package (Table S2). Correlations and differences across conditions were considered statistically significant with $p$ value < 0.05. All statistical analyses were performed in R version 4.3.2.

# Data availability

The compiled dataset of measurements from sampling campaigns and referenced observations of sediment burial and microbial $N_2$ production rates used in this study is available in the Zenodo repository with accession code 18875989.

# Code availability

The R code used to run the spatial bootstrap analyses is in R Markdown format and is available in the Zenodo repository with accession code 18875989.

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

## Acknowledgements

We thank T. Cardis and M. Olsson for their contribution to the sediment sampling in Iceland and Sweden; L. Bristow for advice on water column incubation procedures and for helpful discussions; C.M. Hill for laboratory support; and crews on R/V Skagerak for logistical support. Research funding was provided by the Swedish Research Council VR to S.B. (grant no. 2022-04710). The Iceland and Swedish expeditions were partially funded by the Knut and Alice Wallenberg Foundation (grant no. 2022.0096) and the Swedish Research Council VR (grant no. 2020-00457) to I.R.S., and L.S.L. was supported by the Independent Research Fund Denmark to B.T. (grant no. 1127-00362B).

## Author contributions

H.L.S.C., I.R.S., and S.B. designed the research. H.L.S.C, L.S.L., T.P., I.R.S and S.B. conducted field sampling and collection of geochemical samples. H.L.S.C., L.S.L., and T.P. performed incubation experiments for microbial process rates. C.S. performed analyses on geochemical characteristics and accumulation rates, and provided unpublished geochemical data. H.L.S.C. and L.S.L. performed analyses on GC-IRMS. H.L.S.C. and B.T. conducted formal analysis, compiled geochemical databases, and calculated process rates. S.B. led the research and supervised the project. S.B. and I.R.S. acquired funding for the research. H.L.S.C. wrote the original draft of the manuscript with important contributions from S.B. All authors were responsible for the review and editing of the article.

## Funding

## Competing interests

The authors declare no competing interests.
