## [Peer Review file · Nature Communications]

Long-term nitrogen burial exceeds denitrification in global fjords

Corresponding Author: Mr Henry Cheung

Version 0:

Reviewer comments:

Reviewer #1

(Remarks to the Author)

This is a well-written, interesting and important paper, which I believe may be accepted for publication after minor edits. I made some comments and proposed some changes directly in the manuscript and supplementary materials word files. The paper is easy to read and conveys the main messages in a very clear manner.

What are the noteworthy results? Revealing the importance of nitrogen burial in fjord systems worldwide and how its relative importance relates with the presence of anoxic environments, where denitrification and anammox gain importance.

Will the work be of significance to the field and related fields? How does it compare to the established literature? If the work is not original, please provide relevant references.

I think so, and it looks to be original.

Does the work support the conclusions and claims, or is additional evidence needed?

I believe that the conclusions are well grounded. Additional evidence is always welcome but a considerable amount of data was used to support the claims made by the authors, covering fjords in both hemispheres and at various latitudes.

Are there any flaws in the data analysis, interpretation and conclusions? Do these prohibit publication or require revision?

I did not detect any fails that should prevent publication or require revision. I found some statements that I believe should be revised but that do not compromise the claims made by the authors. I added comments to the manuscript about these small things that should be corrected.

Is the methodology sound? Does the work meet the expected standards in your field?

I believe so.

Is there enough detail provided in the methods for the work to be reproduced?

Yes. Moreover, the authors provide many methodological details in the Supplementary material.

Reviewer #2

(Remarks to the Author)

General assessment

This review concerns the manuscript "Long term nitrogen burial exceeds denitrification in global fjords" by Cheung, Levin, Smeaton, Politi, Thamdrup, Santos, and Bonaglia. I would like to recognize the considerable effort invested in this study, particularly the integration of multiple techniques, extensive literature review, and conceptual development. The work addresses an important gap in nitrogen cycling in fjords and provides valuable insights. The comments below reflect my personal opinion and are intended to improve the clarity, robustness, and relevance of the manuscript.

This manuscript addresses an important gap in global nitrogen cycling by quantifying burial versus microbial N loss in fjords. The topic is novel and relevant, and the integration of new measurements with a global synthesis is a major strength. The methods (radiotracer dating, isotope pairing, spatial bootstrap) are technically sound, and the conceptual model (Fig. 5) is a useful framework. However, several aspects require clarification or improvement to strengthen the manuscript. In its present form, the discussion is clear which makes the key results easy to understand. However, some of the interpretations remain "too obvious" and I think the manuscript will benefit a more thorough discussion of the results to provide more critical findings. I also have concerns with the statistical analyses.

Below are more detailed comments:

- Abstract

L17–18: The statement on long-term sequestration is too categorical. Geological stability can be disrupted by natural hazards (e.g., landslides), which should be acknowledged.

L18–20: The link between accelerated polar warming and increased burial/microbial N loss needs clearer mechanistic explanation.

Main

L29–31: Expand on anthropogenic pressures in mid/high-latitude fjords (e.g., aquaculture, salmon farming, wastewater inputs).

The phrasing of “anthropogenic enrichment” is misleading. Climate-driven processes (glacier melt, soil erosion) are not direct anthropogenic inputs. Consider explicitly distinguishing direct human activities from climate-mediated changes. Add a note on stratification and its role in isolating deep waters, as this is critical for deoxygenation.

Large nitrogen burial rates in global fjords

L75 : This needs a reference (percentage of global marine N burial)

L78–79: Since N burial rates are derived from MAR, the observed correlation seems circular. Perhaps clarify this in the text to avoid misinterpretation as an independent relationship.

Fjords N burial surpasses microbial N loss by denitrification and anammox

(Sometimes you use « nitrogen » and other « N » in title. Please be consistent)

L111-112 : Please clarify the statement on lower OM in arctic fjords

L120 : Please clarify « hypoxic sediment ». Do you mean hypoxic bottom water or sediment layer ?

L122 : Comparing to the St. Lawrence Estuary is problematic, it is not a fjord. Crowe et al. studied the estuary, not Saguenay Fjord in the St Lawrence system. Same for Thibodeau et al. I am not aware of published data on N cycling in the Saguenay Fjord.

Fate of nitrogen in deoxygenated, eutrophic and warmer fjords

L141-148: This paragraph is confusing. It begins by stating that N₂ production is higher in anoxic than in oxic fjords, but then claims that N₂ production is similar to or higher in oxygen-depleted fjords (interpreted here as hypoxic but not fully anoxic) than in the anoxic Baltic Sea. However, the subsequent discussion focuses solely on anoxic fjords.

Could anoxia reduce N₂ production because nitrification cannot occur? This might explain why N₂ production is similar to or higher in oxygen-depleted (hypoxic) than in fully anoxic fjords. In that case, assuming climate change will lead to anoxia (L148), the increase in N₂ production would represent a transient state between oxic and anoxic conditions.

L142 : Please clarify « geological vs biotic dominated N-loss »

L144: Could N₂ production occur in anoxic microenvironments within suspended organic matter? Worth mentioning but from your data it doesn't seem so.

Clarify “displacement” vs. “expansion” of the nitrate reduction zone.

L162-163 : This needs a reference.

Overall, in the present form, the climate projections seem speculative and would need more support to appear robust. Also, the statistical analyses used in this study, while correct, are not the most efficient ones to make robust predictions.

Summary and implication

L168-169 : You mean « currently » ?

L170 : Could you define « blue nitrogen » ?

L170-171 : The percentages do not add up to 100

L171-172 : You mean « N loss » other wise it is somewhat obvious that burial play a key role in long-term N storage.

L174-175 : Nitrification is often the main process producing N₂O

L175-178 : It is not easy for me to follow the rationals here. What comes from your results and from the littérature ?

L185 : You show a clear shift following anoxia, not deoxygenation

L189 : I do not think you defined « eutrophication » in the manuscript

L189-190 : Do you think that enhanced N burial in the sediment could also stimulate CO₂ and CH₄ production which, if released, might offset the reduction in N₂O production?

Methods

Global fjord database collection and upscaling

For your upscaling, mention the assumption that fjord seafloors always consist of soft sediments. This holds for deep basins but not always for sills, steep fjord « walls », or areas with boulders from glacier/sea-ice melt.

Do you have details on sample locations within the fjords for the data you retrieve from literature? High spatial heterogeneity often occurs between the head and mouth.

Statistical analyses

The Spearman rank correlation test assesses monotonic relationships (not necessarily linear), so why is a linear model shown in Figure 3? Moreover, no predictions can be drawn from the Spearman correlation test. Since you show linear models, could you consider using GLMs (or GAMs if not linear) on these data instead, and potentially incorporate multiple predictors simultaneously? This could strengthen your points and broaden the scope of your data.

Given the differences in n between groups (see Figures 2 and 4), could you also report effect sizes?

I think the Wilcoxon rank-sum test is not the best option here because you have three groups. It is like doing pairwise tests without applying a p-value correction, which increases the risk of false positives.

Another point: one group has n = 2, which is too small for reliable statistical testing. It is nice that the data are actually shown

in Figures 2 and 4, but from my perspective the anoxic group should be treated as exploratory observations. You could try permutation tests, although this would not fully resolve the issue of such a small group. As mentioned above, effect sizes may be more meaningful in this case. To be fully transparent about the likely large uncertainty in this group, you could also report bootstrap confidence intervals which could be large.

From my perspective, the unfortunate lack of data from anoxic fjords, combined with the global variability in the dataset, undermines the robustness of the findings linking nitrogen cycling to deoxygenation.

Figures

Make sure to use a color-blind-friendly palette.

Figure 3: I think the figure would benefit from a clearer rationale for how the panels are arranged. At the moment, the sequence does not follow an obvious logic, at least to me. For example, NAR is shown as a function of latitude (a) and of the C:N ratio (d), and then the C:N ratio is shown as a function of latitude (e). What are the objectives of presenting each of these relationships? Are these panels simply a subset of all significant relationships among the possible variable combinations or rely on hypotheses?

Supplementary material

Methods

The methodology is technically sound; a few suggestions:

L91–108: For reproducibility, could you report the $^{15}\text{N-NO}_3$ concentration used for this?

Figures

Figure S3 is not referenced in the main text. I understand the conceptual idea behind the exponential model, but would a segmented model or GAM be more informative? With $n=2$ in anoxia, you may not be able to detect a significant tipping point, but using the segmented or the `mcp` R package within a Bayesian framework might still allow you to explore potential « real » change points.

Does the reported O_2 concentration correspond to bottom-water values, and did you test for a relationship between N burial and O_2 ? Finally, from this figure it is not obvious that N burial exceeds N_2 production, especially under well-oxygenated bottom waters, where most organic matter may already have been remineralized through oxic processes before reaching the seafloor. This point seems worth discussing.

I look forward seeing this work refined.

Version 1:

Reviewer comments:

Reviewer #2

(Remarks to the Author)

General assessment

This review concerns the revised manuscript "Long-term nitrogen burial exceeds denitrification in global fjords" by Cheung, Levin, Smeaton, Politi, Thamdrup, Santos, and Bonaglia. I want to sincerely congratulate the authors for their effort in this revision. I appreciate the care with which they addressed most of my comments and those of the other reviewer. This is a valuable contribution.

Specific comments

Two comments remain on this manuscript:

1- Change points analysis

It would be more transparent to report the full model output, including a summary of posterior distributions (to convey confidence intervals), model diagnostics such as R-hat (convergence diagnostic) and effective sample size, and a clear statement of any priors used.

2- Lower St Lawrence estuary as a fjord-like system

This is the most critical point.

From my perspective, the Lower St. Lawrence Estuary (LSLE) is not a fjord but an estuarine basin. This basin is a rift system subsequently overdeepened and modified by glacial erosion during the last glaciation/deglaciation (Tremblay et al. 2003 : 10.1016/S0040-1951(03)00227-0; Duchesne et al. 2010 : 10.1111/j.1365-2117.2009.00457.x). It is true that a cross-sectional profile of the Laurentian Channel shows a U-shape, with steep walls at some locations (Figure 2 from Normandeau et al. 2015; 10.1016/j.geomorph.2015.03.023). However, the whole LSLE lacks defining characteristics of a fjord. For instance, the LSLE is not narrow, and its walls are not steep. Moreover, the sill is located at the head of the LSLE, near Tadoussac, not at its mouth. This is visible in Figure 3 from Cyr et al. (2015; 10.1002/2014JC010272) cited by the authors, and figure 3 from Bluteau et al. (2021; 10.5194/os-17-1509-2021). I therefore strongly suggest the authors to be extremely cautious when classifying the LSLE as a fjord or fjord-like system, even if this classification has precedent in the literature. The « Fjord-type estuarine » classification is also problematic since the stratification is not due to a sill at the entrance of the Laurentian Channel. Maybe a more descriptive terminology « Laurentian Channel estuarine system » and justify in the method why they consider it similar as a fjord in its biogeochemical functioning. I strongly feel that this is particularly important for a publication in a high impact venue that is likely to be widely cited. Such classifications should be critically reassessed rather than inherited.

RESPONSE TO THE REVIEWERS' COMMENTS

We appreciate the constructive comments provided by the the reviewers. We have addressed all comments below (in blue), alongside the original comments (in black).

Reviewer #1 (Remarks to the Author):

This is a well-written, interesting and important paper, which I believe may be accepted for publication after minor edits. I made some comments and proposed some changes directly in the manuscript and supplementary materials word files. The paper is easy to read and conveys the main messages in a very clear manner.

What are the noteworthy results?

Revealing the importance of nitrogen burial in fjord systems worldwide and how its relative importance relates with the presence of anoxic environments, where denitrification and anammox gain importance.

Will the work be of significance to the field and related fields? How does it compare to the established literature? If the work is not original, please provide relevant references.

I think so, and it looks to be original.

Does the work support the conclusions and claims, or is additional evidence needed?

I believe that the conclusions are well grounded. Additional evidence is always welcome but a considerable amount of data was used to support the claims made by the authors, covering fjords in both hemispheres and at various latitudes.

Are there any flaws in the data analysis, interpretation and conclusions? Do these prohibit publication or require revision?

I did not detect any fails that should prevent publication or require revision. I found some statements that I believe should be revised but that do not compromise the claims made by the authors. I added comments to the manuscript about these small things that should be corrected.

Is the methodology sound? Does the work meet the expected standards in your field?

I believe so.

Is there enough detail provided in the methods for the work to be reproduced?

Yes. Moreover, the authors provide many methodological details in the Supplementary material.

We thank Reviewer #1 for the supportive feedback and for the suggested changes, which positively shaped our revisions and further improved the quality of the manuscript.

Comments on the manuscript

L. 10: *"Global fjord systems account for up to 11% of total marine C sequestration".*

This does not seem correct. Please see my comment below about these numbers. The value of 11% is based on reference number 3, which suggests that fjord system bury 11% of the total buried by marine carbon burial. However, burial is not the only form of sequestration. A large part of sequestration is due to cooling and under saturation of ocean water at high latitudes.

We have edited this passage to be precise and avoid confusion:

"Global fjord systems account for up to 11% of total marine C burial." (Page 1, Line 10).

L. 24: *"fjords are also key coastal marine ecosystems that bury up to 11% of marine carbon globally".*

Please see my comment in the Abstract. I suggest specifying here that is 11% of the total carbon buried in marine systems to avoid confounding with total sequestration.

We have reformulated this sentence to avoid confusion:

“... , fjords are also key coastal marine ecosystems burying ~11% of the global marine organic carbon ^{3,4}.” (Page 2, Lines 24–25).

L. 47: “*Arctic amplification favours deoxygenation and enables atlantification*”.

I do not think that this sentence is accurate. How does Arctic amplification favour Atlantification? I don't recall any study supporting this statement. I would put it the other way around: “Atlantification enhances Arctic amplification”. Amplification likely results from the climate-albedo feedback but areas more exposed to Atlantification are those warming faster in the Arctic.

We agree and have revised the statement with new references to avoid confusion:

“Arctic atlantification, i.e., the extension of warm, nutrient-rich Atlantic waters into the Arctic ²⁶, amplifies warming in the Arctic Ocean ²⁷, favouring deoxygenation ²⁸ and thereby altering microbial N transformation ²⁹.” (Page 3, Lines 49–51).

L. 96: “*primary production sustained by glacier meltwater nutrients*”.

I don't think one may generalize that primary production is sustained by glacier meltwater nutrients. In several fjords, nutrient inputs from the sea are the major source and in many cases the subglacial fjord discharges promote upwelling of nutrient-rich fjord deep water, without providing by themselves a significant input of nutrients. See, for example, the following references: 10.1038/S41467-018-05488-8, 10.1038/S41598-025-06953-3

We reformulated the statement as follows:

“This aligns with the low sediment C:N ratios in polar fjords, where organic matter is predominantly derived from marine primary production ⁵. Notably, high productivity can be sustained by nutrient inputs from nutrient-rich deep water upwelling, subglacial discharge, and/or N-rich meltwater from retreating glaciers ^{43,44,45}.” (Page 5, Lines 97–100).

L. 100: “*This burial is expected to increase, giving fjords a central role in global N cycle*”.

Why is it expected to increase?

We have incorporated a predictive model for N burial, which includes sediment C:N ratio (i.e., lability of deposited organic matter) and sedimentation rates as key predictors (Fig. 3a). Considering their stimulatory effects, we reformulated this section to better support this statement:

“High sedimentation rates shorten O₂ exposure for organic matter degradation, allowing efficient preservation of labile organic matter ⁴⁶ and associated N burial in Arctic fjords ⁴⁷. Combined with high sedimentation rates from retreating calving glaciers ⁴⁸, an even higher N burial is currently occurring in high-latitude fjords. With increasing sedimentation and decreasing organic matter C:N ratio (Fig. 3a), N burial in fjords is also expected to increase.” (Page 5, Lines 100–104).

Figure 3 Predictive models for sediment nitrogen burial and N₂ production across fjords globally. (a) Observed versus predicted fjord nitrogen accumulation rates (N_{AR}) derived from a generalized linear model including sediment mass accumulation rates (M_{AR}) and sediment C:N ratio (C:N). (b) Observed versus predicted fjord N₂ production rates derived from a generalized linear model including water temperature (temp), nitrate and nitrite concentrations (NO_x), and restricted maximum likelihood fitted oxygen concentrations (O_2). Data points are colour-coded on the basis of fjord regions and the dashed line indicates the 1:1 relationship. Model coefficients, goodness-of-fit (R^2) and effective degrees of freedom for the O_2 fit ($edf_{f(O_2)}$) are shown.

L. 156–158: “Ocean warming also stimulates atlantification – the northward inflow of warmer, nutrient-rich waters from the Atlantic into the Arctic”.

As I mentioned above, I never saw any clear demonstration that Atlantification is stimulated by ocean warming. May the authors clarify the statement? I understand that the effects of Atlantification will be stronger under ocean warming since Atlantic water will be warmer, but I am not sure that the inflow of Atlantic water changes because of ocean warming. I never saw any evidence for that.

We thank the reviewer for pointing out this issue. To clarify, we have reformulated the statement and added references as follows:

“Ocean warming also promotes the northward expansion of warmer, nutrient-rich waters from the Atlantic into the Arctic^{61, 62, 63}.” (Page 8, Lines 168–170).

L. 160–162: “However, these microbial feedbacks may occur more slowly due to the resistance of sediments to temperature shifts, and sediment N₂ production might not always be limited by inorganic N availability”.

I may have misunderstood the sentence, but it looks like the first part challenges the increase in microbiological N₂ removal due to sediment temperature shifts, whereas the second seems to argue in the opposite direction, claiming that N₂ production might not be limited by inorganic N availability. May you please clarify?

We have reformulated this statement to improve clarity:

“In addition to the direct stimulatory effects of warming and nutrient enrichment onto N₂ production (Fig. 3b), the increased availability of labile organic matter due to atlantification-driven primary production may increase sediment N₂ production in Arctic fjords^{45, 64, 67}.” (Page 8, Lines 172–175).

L. 194: Section – Global fjord database collection and upscaling. I suggest removing details about the calculation of N burial and N₂ production from this paragraph, since such details are provided below in a dedicated paragraph. Here you may just list the geographic details of the data and upscaling.

We have minimised details about N burial as suggested by the reviewer. Instead of describing the procedure to calculate N₂ production rates provided in another paragraph, this section now only describes the methodological criteria for data collection in the literature to avoid methodological artefacts (Eyre et al., 2002).

Comments on Supplementary Materials

L. 33: “*Small river inputs from the heads of both fjords regarding the drainage basins <160 km²*”. Please rephrase for clarity

We have reformulated the sentence to improve clarity:

“The relatively small drainage basins (<160 km²) resulted in limited river inputs to both fjords.”
(Page 2, Lines 33–34).

Reviewer #2 (Remarks to the Author):

General assessment

This review concerns the manuscript “Long term nitrogen burial exceeds denitrification in global fjords” by Cheung, Levin, Smeaton, Politi, Thamdrup, Santos, and Bonaglia. I would like to recognize the considerable effort invested in this study, particularly the integration of multiple techniques, extensive literature review, and conceptual development. The work addresses an important gap in nitrogen cycling in fjords and provides valuable insights. The comments below reflect my personal opinion and are intended to improve the clarity, robustness, and relevance of the manuscript.

This manuscript addresses an important gap in global nitrogen cycling by quantifying burial versus microbial N loss in fjords. The topic is novel and relevant, and the integration of new measurements with a global synthesis is a major strength. The methods (radiotracer dating, isotope pairing, spatial bootstrap) are technically sound, and the conceptual model (Fig. 5) is a useful framework. However, several aspects require clarification or improvement to strengthen the manuscript. In its present form, the discussion is clear which makes the key results easy to understand. However, some of the interpretations remain “too obvious” and I think the manuscript will benefit a more thorough discussion of the results to provide more critical findings. I also have concerns with the statistical analyses.

We thank Reviewer #2 for their constructive feedback and the encouraging tone. We have carefully addressed all points raised by the reviewer in our responses below.

Below are more detailed comments:

- Abstract

L17–18: The statement on long-term sequestration is too categorical. Geological stability can be disrupted by natural hazards (e.g., landslides), which should be acknowledged

We have also acknowledged the influence of natural hazards as follow:

“However, natural hazards such as landslides and floods can alter sediment delivery and hence N burial efficiency^{71, 72}.” (Page 9, Lines 193–194).

L18–20: The link between accelerated polar warming and increased burial/microbial N loss needs clearer mechanistic explanation.

We have reformulated the sentence to explain the mechanism while staying within the abstract word limit:

“Accelerated warming in polar regions will promote both N burial from increased primary production and microbial N loss from warmer temperatures, affecting the N budgets in fjords and in the ocean in general.” (Page 1, Lines 18–20).

Main

L29–31: Expand on anthropogenic pressures in mid/high-latitude fjords (e.g., aquaculture, salmon farming, wastewater inputs). The phrasing of “anthropogenic enrichment” is misleading. Climate-driven processes (glacier melt, soil erosion) are not direct anthropogenic inputs. Consider explicitly distinguishing direct human activities from climate-mediated changes.

We have reformulated the sentence as follow:

“Quantification of N loss in mid- to high-latitude blue carbon ecosystems is hence critical to understanding their role in mitigating N enrichment from both direct (agricultural runoff, aquaculture, and wastewater) and indirect (glacial meltwater and soil erosion) sources.” (Page 2, Lines 31–33).

Add a note on stratification and its role in isolating deep waters, as this is critical for deoxygenation.

We have added the following statement to acknowledge the role of stratification in fjord deoxygenation:

“Additionally, warming-induced stratification reduces O₂ supply to fjord deep waters, thereby favouring deoxygenation.” (Page 2, Lines 27–29).

Large nitrogen burial rates in global fjords

L75 : This needs a reference (percentage of global marine N burial)

We have added a reference for global marine N burial (ref. ¹⁷; Zhang et al. (2020)). (Page 4, Line 77).

L78–79: Since N burial rates are derived from MAR, the observed correlation seems circular. Perhaps clarify this in the text to avoid misinterpretation as an independent relationship.

We have reformulated the sentence to avoid misinterpretation:

“Since N burial rates are derived from mass accumulation rates (M_{AR}), the high N burial capacity is necessarily linked to M_{AR} (Fig. 3a).” (Page 4, Lines 80–81).

Fjords N burial surpasses microbial N loss by denitrification and anammox
(Sometimes you use « nitrogen » and other « N » in title. Please be consistent)

We have revised this subtitle for consistency.

L111-112 : Please clarify the statement on lower OM in arctic fjords

We have revised the statement as follow:

“The lower regional median (~0.9 g N m⁻² yr⁻¹) in Greenland and Svalbard likely reflects lower temperatures and organic content availability for N₂ production processes compared to

temperate fjords, given that both temperature and organic carbon were positively correlated with sediment N₂ production (Fig. S1)." (Page 6, Lines 114–117).

L120 : Please clarify « hypoxic sediment ». Do you mean hypoxic bottom water or sediment layer ?

We have reformulated the sentence to improve clarity:

"For instance, greater sediment denitrification and anammox rates were measured when bottom water O₂ concentrations decreased from 260 to 61 μM in Gullmar Fjord, Sweden (Fig. S3)." (Page 6, Lines 126–128).

L122 : Comparing to the St. Lawrence Estuary is problematic, it is not a fjord. Crowe et al. studied the estuary, not Saguenay Fjord in the St Lawrence system. Same for Thibodeau et al. I am not aware of published data on N cycling in the Saguenay Fjord.

Measurements reported in Crowe et al. and Thibodeau et al. were conducted in the Lower St. Lawrence Estuary (LSLE). The LSLE is often defined as a proper fjord system (Bianchi et al., 2020) due to its inherent bathymetric and oceanographic features (El-Sabh & Silverberg, 2012). The LSLE is characterised by a narrow, glaciated submarine valley as a result of glacial erosion (Dyke & Prest, 1987). Alike a typical fjord, the water column in the LSLE is permanently stratified due to the presence of a "sill" (Cyr et al., 2015; Saucier et al., 2003). Because of these reasons and to remain consistent with classifications used in the previous studies, we decided to keep the data from the LSLE. To avoid confusion, we have clarified that those data were originated from measurements conducted in the fjord-like LSLE (Page 6, Line 130) as in the previous review paper (Bianchi et al., 2020):

"Similarly, greater N₂ production was observed in the hypoxic fjord-like Lower St. Lawrence Estuary⁵¹ and Loch Etive⁵² compared to oxic fjords (Fig. S2)." (Page 6, Lines 128–130).

Fate of nitrogen in deoxygenated, eutrophic and warmer fjords

L141-148: This paragraph is confusing. It begins by stating that N₂ production is higher in anoxic than in oxic fjords, but then claims that N₂ production is similar to or higher in oxygen-depleted fjords (interpreted here as hypoxic but not fully anoxic) than in the anoxic Baltic Sea. However, the subsequent discussion focuses solely on anoxic fjords.

We agree that the previous version was confusing. We have now replaced "O₂-depleted" with "anoxic" (Page 7, Line 153). We have also revised the second sentence in the paragraph that now reads:

"Fjord N₂ production increased gradually from oxic to hypoxic conditions with a first threshold at 100 μM O₂, followed by a sharp increase as O₂ declined toward anoxia down to low micromolar range (<20 μM) (Fig. S5)." (Page 7, Line 151–153).

Could anoxia reduce N₂ production because nitrification cannot occur? This might explain why N₂ production is similar to or higher in oxygen-depleted (hypoxic) than in fully anoxic fjords. In that case, assuming climate change will lead to anoxia (L148), the increase in N₂ production would represent a transient state between oxic and anoxic conditions.

We have fixed the previous misunderstanding, i.e., in this context we used "oxygen-depleted" = "anoxic" ≠ "hypoxic". Indeed, N₂ production in anoxic conditions (<20 μM O₂), which exceeded that measured in hypoxic and oxic conditions (Fig. S5), was mainly due to the regime shift and expansion of the nitrate reduction zone, where anoxia persists yet nitrate is still available for N₂ production (Page 6, Lines 133–136).

L142 : Please clarify « geological vs biotic dominated N-loss »

We have reformulated the sentence to improve clarity:

“Up to six-fold greater N₂ production in anoxic than oxic fjords (Fig. 4a) implies that complete deoxygenation shifts the dominant N removal pathway from sedimentary burial toward microbially mediated N loss.” (Page 7, Lines 149–151).

L144: Could N₂ production occur in anoxic microenvironments within suspended organic matter? Worth mentioning but from your data it doesn't seem so.

This is an interesting question. To avoid speculation, we have opted not to comment on anoxic microenvironments because we do not have measurements or literature support to assess the contribution of microenvironments to overall N₂ production in fjords.

Clarify “displacement” vs. “expansion” of the nitrate reduction zone.

We have expanded the sentence as follow:

“The substantial increase in average N₂ production in anoxic fjords is primarily due to the regime shift and expansion of the nitrate reduction zone, i.e., the anoxic but nitrate-containing zone⁵⁹ (Fig. 5).” (Page 7, Lines 155–157).

L162-163 : This needs a reference.

We added ref.⁴⁵, Sørensen et al. (2015). (Page 8, Line 175).

Overall, in the present form, the climate projections seem speculative and would need more support to appear robust. Also, the statistical analyses used in this study, while correct, are not the most efficient ones to make robust predictions.

To improve the robustness of our analyses, we have now incorporated model predictions that include multiple environmental predictors (new Fig. 3) as suggested by the reviewer. We also added new discussion about these findings as follows:

“In addition to the direct stimulatory effects of warming and nutrient enrichment onto N₂ production (Fig. 3b), the increased availability of labile organic matter due to atlantification-driven primary production may increase sediment N₂ production in Arctic fjords^{45, 64, 67}. Climate change stimulates primary production in high-latitude fjords and subsequent deposition of labile organic matter on sediments⁴⁵, favouring both N burial and benthic N₂ production (Fig. S1). Combined with potentially higher N burial due to warming-mediated glacial retreat and increased sedimentation rates⁴⁸, our results suggest an increased N loss in global fjords along with a warmer and less oxygenated ocean.” (Page 8, Lines 172–179).

Summary and implication

L168-169 : You mean « currently » ?

Yes, we have revised the sentence accordingly:

“Our global analysis demonstrates that sediment burial is the most effective N sink mechanism in fjord systems under current conditions.” (Page 9, Lines 181–182).

L170 : Could you define « blue nitrogen » ?

We have added the definition accordingly:

“Despite covering <0.1% of the ocean, fjords contribute up to 18% of global N burial. This makes them hotspots of “blue nitrogen”, i.e., nitrogen sequestered in marine sediments.” (Page 9, Lines 182–184).

L170-171 : The percentages do not add up to 100

Yes, the reported percentages do not sum to 100% because they represent mean relative contributions with associated uncertainty, rather than fractions of a single global total. We have reformulated the sentence to improve clarity:

“On average across fjords globally, sediment burial accounts for $59.4 \pm 4.9\%$ of total N loss, exceeding denitrification ($34.2 \pm 4.2\%$) and anammox ($6.4 \pm 2.6\%$).” (Page 9, Lines 184–185)

L171-172 : You mean « N loss » other wise it is somewhat obvious that burial play a key role in long-term N storage.

Yes, we referred to N removal from the ecosystem. We have reformulated the sentence accordingly:

“Thus, burial plays a surprisingly significant role in the long-term N removal of fjords globally, making it a crucial mechanism for mitigating coastal N enrichment at medium and high latitudes.” (Page 9, Lines 185–187)

L174-175 : Nitrification is often the main process producing N₂O

We agree with the reviewer. We have reformulated the sentence to improve clarity:

“Additionally, N burial is an effective removal pathway with minimum climatic impact, compared to microbial nitrification and denitrification, which produce the greenhouse gas nitrous oxide (N₂O; equivalent to ~1% of N₂ production in an anoxic fjord; ref. ^{21, 68}).” (Page 9, Lines 187–190)

L175-178 : It is not easy for me to follow the rationals here. What comes from your results and from the littérature ?

We agree that the previous text was not easy to follow and we have restructured these sentences that now reads:

“N burial in fjords is driven by the sedimentation rates and higher lability of deposited organic particles (Fig. 3a), which can be favoured by increasing rates of primary production, particle deposition, and glacial retreat ^{69, 70}. However, natural hazards such as landslides and floods can alter sediment delivery and hence N burial efficiency ^{71, 72}” (Page 9, Lines 190–194)

L185 : You show a clear shift following anoxia, not deoxygenation

We reformulated accordingly:

“Indeed, we show that microbial N loss will dominate over geological burial loss due to the shift in N₂ production regimes following development of anoxia (Fig. 5).” (Page 9, Lines 199–201)

L189 : I do not think you defined « eutrophication » inthe manuscript

We added a definition of eutrophication accordingly:

“Whether N burial can counter coastal nutrient overenrichment (i.e., eutrophication) and curb greenhouse gas emissions largely depends on coastal nutrient management.” (Page 10, Lines 205–207)

L189-190 : Do you think that enhanced N burial in the sediment could also stimulate CO₂ and CH₄ production which, if released, might offset the reduction in N₂O production?

We agree and feel that enhanced N burial may potentially stimulate anaerobic sediment processes, potentially affecting the production of all three key greenhouse gases. However, because parallel greenhouse gas flux measurements are unavailable, we cannot evaluate this relationship and would prefer to avoid excessive speculation in the manuscript.

Methods

Global fjord database collection and upscaling

For your upscaling, mention the assumption that fjord seafloors always consist of soft sediments. This holds for deep basins but not always for sills, steep fjord « walls », or areas with boulders from glacier/sea-ice melt. Do you have details on sample locations within the fjords for the data you retrieve from literature? High spatial heterogeneity often occurs between the head and mouth.

In the compiled global dataset, the mean (\pm SD) sediment porosity was 0.78 ± 0.06 ($n = 80$). We added this information accordingly:

“We assumed that sediment particle size was relatively homogenous across sites, given a mean porosity of 0.78 ± 0.06 .” (Page 10, Lines 218–219).

In addition, no observable effect on burial and N₂ production was found. We conclude that the effects of sediment heterogeneity on N loss in fjords are negligible.

Statistical analyses

The Spearman rank correlation test assesses monotonic relationships (not necessarily linear), so why is a linear model shown in Figure 3?

The linear models were used to indicate linearity of the relationship in some cases and, indeed, not all tested datasets showed a linear relationship. We have therefore removed all linear models from the correlation analysis plots (See Fig. S1)

Moreover, no predictions can be drawn from the Spearman correlation test. Since you show linear models, could you consider using GLMs (or GAMs if not linear) on these data instead, and potentially incorporate multiple predictors simultaneously? This could strengthen your points and broaden the scope of your data.

We thank the reviewer for the suggestion to generate better predictive models. We have now included GLM and GAM for N_{AR} and N₂ production to reveal key predictors, respectively, in Fig. 3. These models better highlight that: (1) N burial were driven primarily by sedimentation rate (M_{AR}) and lability of buried material (sediment C:N ratio); (2) N₂ production were mainly driven by water temperature, nitrate concentration, and oxygen concentration. These models particularly consider the non-linear effect of O₂ on N₂ production.

Figure 3 Predictive models for sediment nitrogen burial and N_2 production across fjords globally. (a) Observed versus predicted fjord nitrogen accumulation rates (N_{AR}) derived from a generalized linear model including sediment mass accumulation rates (M_{AR}) and sediment C:N ratio (C:N). (b) Observed versus predicted fjord N_2 production rates derived from a generalized linear model including water temperature (temp), nitrate and nitrite concentrations (NO_x), and restricted maximum likelihood fitted oxygen concentrations (O_2). Data points are colour-coded on the basis of fjord regions and the dashed line indicates the 1:1 relationship. Model coefficients, goodness-of-fit (R^2) and effective degrees of freedom for the O_2 fit ($edf_{(O_2)}$) are shown.

Given the differences in n between groups (see Figures 2 and 4), could you also report effect sizes? I think the Wilcoxon rank-sum test is not the best option here because you have three groups. It is like doing pairwise tests without applying a p-value correction, which increases the risk of false positives. Another point: one group has $n = 2$, which is too small for reliable statistical testing. It is nice that the data are actually shown in Figures 2 and 4, but from my perspective the anoxic group should be treated as exploratory observations. You could try permutation tests, although this would not fully resolve the issue of such a small group. As mentioned above, effect sizes may be more meaningful in this case. To be fully transparent about the likely large uncertainty in this group, you could also report bootstrap confidence intervals which could be large. From my perspective, the unfortunate lack of data from anoxic fjords, combined with the global variability in the dataset, undermines the robustness of the findings linking nitrogen cycling to deoxygenation.

We have now used a one-way permutation test to assess differences among groups. Effect sizes, as the reviewer suggested, are now also reported for all tests (See revised Fig. 2 and 4 below). For Fig. 4, we removed pairwise comparisons between groups to avoid misinterpretation of the exploratory statistical analysis.

About the lack of data from anoxic fjords: Rates from anoxic fjords were comparable to those in the anoxic, brackish Baltic Sea, supporting their reliability. In addition, the low number of observations reflects the fact that, under current conditions, anoxic fjords are generally quite rare. Because of these reasons, we have decided to keep the anoxic group for the global analysis, yet interpret it cautiously. We explicitly acknowledge these limitations in the text:

“While based on measurements from only two anoxic fjords, N_2 production was notably higher under anoxic conditions (Fig. 4a).” (Page 6, Lines 119–120)

Bootstrap CIs of the group were large, and values were very close to the range of the two data points in anoxic fjords. To avoid potential misinterpretation, we hence chose not to report these values.

Figure 2 Distribution of accumulation rates and stoichiometry in fjord sediments across ocean basins. The range and variability of sediment (a) nitrogen accumulation rates (N_{AR}), (b) mass accumulation rates (M_{AR}), and (c) sediment carbon to nitrogen molar ratio (C:N) across ocean basins (colour-coded). Dashed horizontal line indicates bootstrapped global median value. Crosses indicate bootstrapped medians of the corresponding region. Differences among ocean basins were assessed using permutation-based one-way tests, with global p -values reported. Effect sizes were quantified using epsilon-squared (ϵ^2) derived from the Kruskal–Wallis statistic.

Figure 4 Rates of N loss in fjords under different redox conditions. (a) N_2 production and (b) nitrogen accumulation rates (N_{AR}) in anoxic, hypoxic, and oxic fjords. (c) contribution of N burial to total fjord N loss (sum of N_2 production and N burial) under each redox condition. Global differences among redox conditions were assessed using permutation-based one-way tests, with p -values reported. Effect sizes were quantified using epsilon-squared (ϵ^2). Pairwise differences relative to oxic conditions were evaluated using permutation-based post hoc tests with Holm correction, with adjusted p -values shown.

Figures

Make sure to use a color-blind-friendly palette.

We have revised all plots and have used color-blind-friendly palette accordingly.

Figure 3: I think the figure would benefit from a clearer rationale for how the panels are arranged. At the moment, the sequence does not follow an obvious logic, at least to me. For example, N_{AR} is shown as a function of latitude (a) and of the C:N ratio (d), and then the C:N ratio is shown as a function of latitude (e). What are the objectives of presenting each of these relationships? Are these panels simply a subset of all significant relationships among the possible variable combinations or rely on hypotheses?

We have now included only predictive models of N_{AR} and N_2 production in revised Figure 3. For other relationships presented in the previous version, we have added supplementary Fig. S1 shown below, including: (1) Latitudinal trends of N burial, sediment C:N ratio and fjord N_2 production; (2) Environmental controls (C burial, C:N ratio, and bottom water O_2) on N burial; (3) Environmental controls (bottom water temperature, sediment organic carbon content, and bottom water nitrate+nitrite concentration) on sediment N_2 production (water column N_2 production excluded).

Figure S1 Relationship between fjord N loss process rate and environmental factors. **(a-c)** Latitudinal trends in fjord nitrogen burial (N_{AR}), sediment C:N ratio, and fjord N_2 production across fjord sites, respectively. **(d-f)** Environmental controls on fjord NAR, including organic carbon accumulation rates (OC_{AR}), sediment C:N ratio, and bottom water oxygen (O_2) concentration, respectively. **(g-i)** Environmental controls on fjord sediment N_2 production rate (water column production excluded), including bottom water temperature, sediment organic carbon content (OC), and bottom water nitrate + nitrite concentration, respectively. Data points are colour-coded on the basis of fjord regions. Spearman correlation coefficient (ρ_s) and corresponding p-value are shown.

Supplementary material

Methods

The methodology is technically sound; a few suggestions:

L91–108: For reproducibility, could you report the $^{15}\text{N}\text{-NO}_3^-$ concentration used for this?

The $^{15}\text{N}\text{-NO}_3^-$ concentrations were reported in the main text Methods:

“Collected intact cores were spiked with $^{15}\text{NO}_3^-$ (11, 24, 56, and 120 μM mean final concentration) ...” (Page 12, Lines 261–262)

Figures

Figure S3 is not referenced in the main text. I understand the conceptual idea behind the exponential model, but would a segmented model or GAM be more informative? With $n=2$ in anoxia, you may not be able to detect a significant tipping point, but using the segmented or the mcp R package within a Bayesian framework might still allow you to explore potential « real » change points.

We thank the reviewer for the suggestion and have adopted the MCP model within our analyses. We yielded two change points of N_2 production rates at 9.1 and 100.4 μM O_2 . This aligns well with our discussion on hypoxia and anoxia in the fjord, which stimulates N_2 production rates. We added the results of the MCP model in the text as follows:

“Fjord N_2 production increased gradually from oxic to hypoxic conditions with a first threshold at 100 μM O_2 , followed by a sharp increase as O_2 declined toward anoxia down to low micromolar range (<20 μM) (Fig. S5).” (Page 7, Line 151–153).

Figure S5 Potential changes in fjord N_2 production as a function of oxygen concentrations. Sediment and water column total N_2 production rates (denitrification + anammox) in global fjords are based on a compilation of measurements from the literature and new data from Sweden and Iceland. A Bayesian segmented regression model was fitted to the observed N_2 production rates. Dashed lines indicate the estimated change points (cp1 and cp2) of shifts in N_2 production rate. An asterisk denotes a higher benthic N_2 production rate reported in Loch Etive under oxygenated conditions, yet this system is known to experience periodic hypoxia¹⁹.

Does the reported O_2 concentration correspond to bottom-water values, and did you test for a relationship between N burial and O_2 ?

Yes, the reported O_2 concentrations were bottom-water values. In addition to no observable differences in N burial rates among redox conditions (Fig. 4b), we tested N burial as a function of bottom-water O_2 concentrations (Fig. S1), with Spearman's $\rho = 0.12$ and $p = 0.52$, suggesting a negligible effect of O_2 concentrations on N burial rates.

Finally, from this figure it is not obvious that N burial exceeds N₂ production, especially under well-oxygenated bottom waters, where most organic matter may already have been remineralized through oxic processes before reaching the seafloor. This point seems worth discussing.

Most of the burial literature data does not report bottom water O₂ concentration. Only 32 out of 163 observations reported bottom water O₂ concentrations during sampling. Therefore, without proper *in situ* O₂ data, we categorised fjords according to their prevalent redox condition (Fig. 4b). We therefore removed the N burial data from this figure (new Fig. S5) to avoid misinterpretation.

I look forward seeing this work refined.

Once again, we thank the editor and two reviewers for their constructive feedback. We feel that this revised version has substantially improved.

END OF REVIEWS

REFERENCES CITED IN THIS RESPONSE LETTER:

- Bianchi, T. S., Arndt, S., Austin, W. E. N., Benn, D. I., Bertrand, S., Cui, X., et al. (2020). Fjords as Aquatic Critical Zones (ACZs). *Earth-Science Reviews*, 203, 103145.
<https://www.sciencedirect.com/science/article/pii/S0012825219305847>
- Cyr, F., Bourgault, D., Galbraith, P. S., & Gosselin, M. (2015). Turbulent nitrate fluxes in the Lower St. Lawrence Estuary, Canada. *Journal of Geophysical Research: Oceans*, 120(3), 2308-2330.
<https://doi.org/10.1002/2014JC010272>
- Dyke, A., & Prest, V. (1987). Late Wisconsinan and Holocene History of the Laurentide Ice Sheet. *Géographie physique et Quaternaire*, 41(2), 237-263.
<https://id.erudit.org/iderudit/032681ar>
- El-Sabh, M. I., & Silverberg, N. (2012). *Oceanography of a large-scale estuarine system: the St. Lawrence* (Vol. 39): Springer Science & Business Media.
- Eyre, B. D., Rysgaard, S., Dalsgaard, T., & Christensen, P. B. (2002). Comparison of isotope pairing and N₂:Ar methods for measuring sediment denitrification—Assumption, modifications, and implications. *Estuaries*, 25(6), 1077-1087.
<https://doi.org/10.1007/BF02692205>
- Saucier, F. J., Roy, F., Gilbert, D., Pellerin, P., & Ritchie, H. (2003). Modeling the formation and circulation processes of water masses and sea ice in the Gulf of St. Lawrence, Canada. *Journal of Geophysical Research: Oceans*, 108(C8). <https://doi.org/10.1029/2000JC000686>
- Sørensen, H. L., Meire, L., Juul-Pedersen, T., de Stigter, H. C., Meysman, F. J. R., Rysgaard, S., et al. (2015). Seasonal carbon cycling in a Greenlandic fjord: an integrated pelagic and benthic study. *Marine Ecology Progress Series*, 539, 1-17. <https://www.int-res.com/abstracts/meps/v539/meps11503>
- Zhang, X., Ward, B. B., & Sigman, D. M. (2020). Global Nitrogen Cycle: Critical Enzymes, Organisms, and Processes for Nitrogen Budgets and Dynamics. *Chemical Reviews*, 120(12), 5308-5351.
<https://doi.org/10.1021/acs.chemrev.9b00613>